# Systematic review and meta-analysis of the effect of ABO blood group on the risk of SARS-CoV-2 infection

**George Balaouras[1], Paolo Eusebi [2], Polychronis Kostoulas [1]** *

**1** Faculty of Public Health, University of Thessaly, Volos, Greece, **2** Department of Medicine and Surgery, University of Perugia, Perugia, Italy

* pkost@uth.gr

**Editor:** Kovy Arteaga-Livias, Hermilio Valdizán National University Academic Professional School of Medicine: Universidad Nacional Hermilio Valdizan Escuela Academico Profesional de Medicina Humana, PERU

**Data Availability Statement:** All data appear in Table 2.

## Abstract

We have been experiencing a global pandemic with baleful consequences for mankind, since the Severe Acute Respiratory Syndrome Coronavirus 2 (SARS-CoV-2) was first identified in Wuhan of China, in December 2019. So far, several potential risk factors for SARS-CoV-2 infection have been identified. Among them, the role of ABO blood group polymorphisms has been studied with results that are still unclear. The aim of this study was to collect and meta-analyze available studies on the relationship between SARS-CoV-2 infection and different blood groups, as well as Rhesus state. We performed a systematic search on PubMed/MEDLINE and Scopus databases for published articles and preprints. Twenty-two studies, after the removal of duplicates, met the inclusion criteria for meta-analysis with ten of them also including information on Rhesus factor. The odds ratios (OR) and 95% confidence intervals (CI) were calculated for the extracted data. Random-effects models were used to obtain the overall pooled ORs. Publication bias and sensitivity analysis were also performed. Our results indicate that blood groups A, B and AB have a higher risk for COVID-19 infection compared to blood group O, which appears to have a protective effect: **(i) A group vs O (OR = 1.29, 95% Confidence Interval: 1.15 to 1.44), (ii) B vs O (OR = 1.15, 95% CI 1.06 to 1.25), and (iii) AB vs. O (OR = 1.32, 95% CI 1.10 to 1.57)**. An association between Rhesus state and COVID-19 infection could not be established **(Rh+ vs Rh- OR = 0.97, 95% CI 0.83 to 1.13)**.

## Introduction

Coronaviruses (COVs) are enveloped viruses with a single positive-stranded RNA genome. They belong to the subfamily Orthocoronavirinae under the family Coronaviridae and are classified into four genera: Alphacoronaviruses (α), Betacoronaviruses (β), Gammacoronaviruses (γ) and Deltacoronaviruses (δ). The viral genome normally encodes four structural proteins, spike (S), envelope (E), membrane (M), and nucleocapsid (N) [1]. The term *coronavirus* refers to the appearance of CoV visions, when observed under electron microscopy, in which spike projections from the virus membrane, give the semblance of a crown, or corona in Latin [2]. To date, seven human CoVs (HCoVs) are known. Among them, HCoV-229E and

**Funding:** This work has been funded under the H2020 project: "Unravelling Data for Rapid Evidence-Based Response to COVID-19". I hereby declare that the funders had had no role in study design, data collection and analysis, decision to publish, or preparation of the manuscript.

**Competing interests:** The authors have declared that no competing interests exist.

HCoV-NL63 are alpha-CoVs. The other five beta-CoVs include HCoV-OC43, HCoV-HKU1, Severe Acute Respiratory Syndrome Coronavirus (SARS-CoV), Middle East Respiratory Syndrome Coronavirus (MERS-CoV) and Severe Acute Respiratory Syndrome Coronavirus 2 (SARS-CoV-2) [3]. In December 2019, a human outbreak of pneumonia, later named coronavirus disease (COVID-19), began spreading across the planet, infecting millions. The causative agent of COVID-19 was quickly identified as a novel coronavirus, the Severe Acute Respiratory Syndrome Coronavirus 2 (SARS-CoV-2). Although close evolutionary relationships to bat CoVs suggest a bat origin for SARS-CoV-2, our understanding is notably limited by the scarcity of available sequenced CoV genome [4]. As a novel beta coronavirus, SARS-CoV-2 shares 79% genome sequence identity with SARS-CoV and 50% with MERS-CoV. Its genome organization is shared with other beta coronaviruses [5].

The spike protein S appears to be critical for cellular entry because it guides the virus to attach to the host cell. The receptor-binding domain (RBD) of the spike protein S binds to Angiotensin-Converting Enzyme 2 (ACE2) to initiate cellular entry [6]. The SARS-CoV-2 virus typically causes respiratory and gastrointestinal sickness. It can be transmitted through aerosols and direct or indirect contact, as well as during medical cases and laboratory sample handling. The disease is characterized by symptoms such as high fever, chills, cough, breathing difficulty, diarrhea, myalgia, fatigue and may occasionally lead to complications like pneumonia, severe acute respiratory syndrome (SARS) and eventually death [7].

After the ABO blood group system was found by Karl Landsteiner in 1901, the search for the relationship between blood groups and various diseases has continued uninterrupted [8]. Recently, several studies have reported an association between blood group and SARS-CoV-2 infection. However, results are conflicting, perhaps due to the potential effect of multiple confounding effects, and controversy remains with respect to the role of blood type on COVID-19 infection [9]. We performed a meta-analysis to assess the association between ABO blood groups, Rhesus state and COVID-19 infection.

## Materials and methods

### Search strategy

A systematic online search for published literature was carried out in PubMed/MEDLINE and Scopus databases, including unpublished articles, with the MESH (medical subject heading) terms "ABO blood groups" and "COVID-19". In order to expand our search scale, we also conducted a full-text search with the relevant terms ("SARS-CoV-2 infection", "2019-nCoV infection", "novel coronavirus infection" and "ABO polymorphisms"). **The searching time period was until March 7[th] 2021** and we limited the search language to English, with no restrictions on country or publication state.

### Study selection

We included the studies that fulfilled the following inclusion criteria: i) studies that reported an association between COVID-19 infection and ABO blood groups and/or Rhesus state; ii) case-control and cohort studies; iii) provision of original data. Excluded studies included: (i) reviews, clinical guidelines, and expert consensus; (ii) animal or in vitro cell studies; (iii) studies for which the full text was not available; (iv) studies with insufficient data.

### Data extraction

Data extraction included: first author's name, publication year, title and the link of the study, case definition, the distribution numbers of participants for each blood group (along with

Rhesus state, when there was a record) and for both, SARS-CoV-2 infected and uninfected subjects. For each study, a numerical ID was used. Infection was confirmed by Polymerase Chain Reaction (PCR) and/or clinical diagnosis, although for several studies the confirmation method for SARS-CoV-2 infection was not specified. Some studies included more than one group of controls, along with the corresponding population of cases, while other studies reported more than one group of controls and cases. We included in the analysis all the comparisons regarding different subgroups of controls and cases, in order to avoid any overlapping.

## Statistical analysis

For each study, we extracted the cross-classified frequencies between infection state and blood group. We used logistic regression for deriving Odds Ratios (ORs) and their asymptotic standard errors, after adjusting for multiplicity using the Benjamin-Hochberg procedure [10]. We assessed heterogeneity using the I-squared statistic. Publication bias was assessed by visual inspection of the funnel plots and further validated by Egger's test [11]. Pooled ORs estimates and 95% confidence intervals (CIs) were obtained by performing meta-analysis using the inverse variance method. Due to the amount of heterogeneity a random-effects model has been used for the ABO gene, by applying the Hartung-Knapp-Sidik-Jonkman method [12] for $\tau^2$. The 95% prediction intervals (PIs) were also computed. The PIs present the heterogeneity in the same metric as the original effect size measure, illustrating which range of true effects can be expected in future settings [13]. We explored the robustness of our meta-analysis results using the leave-one-out method.

## Software

All models were run in R v4.0 using the meta package [14].

## Results

### Literature search

The literature search of the PubMed/MEDLINE and Scopus databases resulted in 589 potentially relevant studies (PubMed records = 389 and Scopus records = 200). The 351 of them were removed because they were duplicates. According to the inclusion criteria, we excluded the 216 irrelevant studies by screening abstract and title. Eventually, a total of 22 articles [15–36] were included in this systematic review and meta-analysis (Fig 1).

### Study characteristics

Twenty-two studies were identified, meeting our inclusion criteria for meta-analysis, with the majority of them being case-control studies. All studies were published in 2020, except for five studies that were published in 2021. Half of the studies were carried out in Europe and North America while the other half in Asia and Africa. A total of 84,659,546 subjects were included in this meta-analysis, with 21,462 COVID-19 infected subjects and 84,638,084 uninfected subjects. Among them, 147,302 subjects were positive for Rhesus state and 20,313 negative. Most of the participants were adult males, forty to seventy years old. In most of the studies, COVID-19 diagnosis was confirmed by a PCR test, using nasal or pharyngeal swab specimens. The main characteristics of the studies are listed in Table 1.

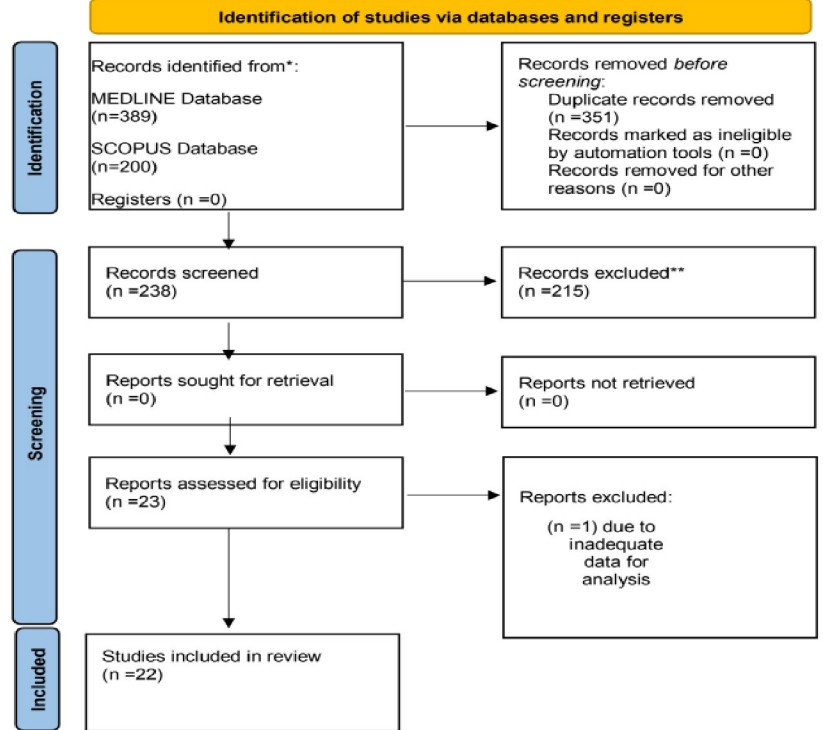

**Fig 1. The PRISMA flow-chart.**

## Characteristics of the included studies

**Association between blood groups and COVID-19 infection.** Meta-analysis for the ABO group (Table 2 and Figs 2–7), revealed increased odds of COVID-19 infection in the (i) A group vs O (OR = 1.29, 95% Confidence Interval: 1.15 to 1.44), (ii) B vs O (OR = 1.15, 95% CI 1.06 to 1.25), and (iii) AB vs. O (OR = 1.32, 95% CI 1.10 to 1.57). Prediction intervals include the reference value of 1 for the OR in all pairwise comparisons. The visual inspection of the funnel plots (Fig 8) and the results of Egger's test showed some evidence of publication bias for the comparison between of A vs. O (p = 0.013) and A vs. B (p = 0.047). Sensitivity analysis by the leave-one-out method provided similar estimates (Supplementary Files).

 **Association between Rhesus status and COVID-19 infection.** Meta-analysis of the association between Rhesus state and COVID-19 infection (Figs 9 and 10) in the 10 studies that included information on Rhesus, did not provide evidence of association with the COVID-19 infection (Rh+ vs Rh- OR = 0.97, 95% CI 0.83 to 1.13). The 95% PI includes the reference value of 1 for the OR in all pairwise comparisons. The leave-one-out sensitivity analysis provided similar estimates (Supplementary Files). Visual inspection of the funnel plot (Fig 5) and the results of Egger's test (p = 0.618) showed no evidence of publication bias.

**Table 1. The main characteristics of the studies.**

| Study Year | Country | Study Design | Sample Size (case/control) | Rhesus Status (positive/negative) | Age. years | Male% (Case/Control) | Patients | Controls |
|---|---|---|---|---|---|---|---|---|
| Boudin et al, 2020 | France | Retrospective Cohort | 1263/406 | 1439/230 | Median Age (IQR): 28(23–36)/27(23–33) | 87/87 | Patients with COVID-19 confirmed by RT-PCR and clinical symptoms suggestive to covid-19 | Tested negative for COVID-19 or no clinical symptoms |
| Fan et al. 2020 | China | Retrospective Case-Control | 105/103 | ND | Mean Age ±SD: (56.8 ±18.3)/(54.0 ±15.0) | 52.4/54.4 | Patients with COVID-19 confirmed by RT-PCR and clinically diagnostic cases | Tested negative for COVID-19 or no clinical symptoms |
| Abdollahi et al. 2020 | Iran | Cross-Sectional | 397/500 | 802/95 | Mean Age (SD): 58.81 (15.4)/48.53 (17.9) | 63.5/46.2 | Patients with COVID-19 confirmed by RT-PCR | Healthy population |
| Rahim et al. 2021 | Pakistan | Cross-Sectional | 1935/1935 | ND | Mean Age ±SD: (39.73 ±15.26)/(32.36 ±8.65) | 68.6/67.7 | Patients with COVID-19 confirmed by RT-PCR | Healthy blood donors |
| Bhandari et al. 2020 | USA | Retrospective Case-Control | 825/396 | 1160/61 | Mean Age ±SD: (57.64 ±18.17)/(54.21 ±20.99) | 61/44 | Patients with COVID-19 confirmed by RT-PCR | Patients who were hospitalized without COVID-19 |
| Barnkob et al. 2020 | Denmark | Retrospective Cohort | 7422/466232 7422/2204742 | ND | Median Age (IQR): 52 (40–67)/50 (36–64) | 32.9/32 | Patients with COVID-19 confirmed by RT-PCR | Tested negative for COVID-19/ Healthy population |
| Kibler et al. 2020 | France | Retrospective Cohort | 22/680 | 352/350 | Mean Age ±SD: (82±8.4)/(82±6.9) | 31.8/45 | Patients with COVID-19 confirmed by RT-PCR and typical symptoms and characteristic imaging findings on chest computed tomography (CT) | Patients who were hospitalized without COVID-19 |
| Muniz-Diaz et al. 2021 | Spain | Retrospective Cohort | 854/75870 965/52584 | ND | Median Age (IQR): 45.0 (36.0–53.0)/ 45.0 (32.0–53.0) | 39.5/51.5 59.07/ 49.85 | COVID-19 blood donors confirmed by RT-PCR /transfused patients with COVID-19 | Healthy blood donors/Patients transfused without COVID-19 |
| Valenti et al. 2020 | Italy | Case-Control | 505/890 505/18097 | ND | Median Age (IQR): 69.0 (59.0–77.0)/ 72.1 (58.2–82.5) | ND | COVID-19 patients.SARS-CoV-2 viral RNA polymerase-chain-reaction (PCR) test from nasopharyngeal swabs or other relevant biologic fluids | Healthy blood donors/transfused patients |
| El-Shitany et al. 2021 | Saudi Arabia and Egypt | Retrospective Cross-Sectional | 726/707 | 1185/248 | ND | 15.2/16.5 | COVID-19 recovered patients. confirmed by RT-PCR and biochemical and clinical symptoms | Healthy population |
| Khalil et al. 2020 | Lebanon | Retrospective Case-Control | 146/6479 | ND | Mean Age ±SD. (IQR): (41.9±18.52). (28–57) CO | 66.4 CO | Patients with COVID-19 confirmed by RT-PCR | Patients who were hospitalized without COVID-19 |
| Wu et al. 2020 | China | Retrospective Case-Control | 187/1991 | ND | ≥40: 63.1% CO | 51.9 CO | Electronic medical records of patients with COVID-19 | Patients who were hospitalized without COVID-19 |
| Gamal et al. 2021 | Italy | Retrospective Case-Control | 1600/27715 | 25206/4104 | ND | ND | Patients with COVID-19 confirmed by RT-PCR | Healthy blood donors |
| Franchini et al. 2021 | Italy | Case-Control | 447/16911 | ND | Mean Age ±SD: (477 ±121)/(471 ±143) | 86.1/61.0 | Blood donors clinically recovered from COVID-19 (SARS-CoV-2 RT-PCR nasal swabs and clinically) | Healthy blood donors |

*(Continued)*

**Table 1.** (Continued)

| Study Year | Country | Study Design | Sample Size (case/ control) | Rhesus Status (positive/ negative) | Age. years | Male% (Case/ Control) | Patients | Controls |
|---|---|---|---|---|---|---|---|---|
| Chegni et al. 2020 | Iran | Case-Control | 76/ 80982137 | ND | >59: 53.2% CO | 77.7 CO | COVID-19 patients. confirmation method was not specified | Healthy population |
| Zalba-Marcos et al. 2020 | Spain | Retrospective Cohort | 225/182384 | ND | Mean Age (SD) of 44% 70.1 (15.1) CO | 64 CO | Patients with COVID-19 confirmed by RT-PCR | Healthy population |
| Dzik et al. 2020 | USA | Case-Control | 957/5840 | ND | ND | ND | Patients with COVID-19 confirmed by RT-PCR | Patients who were hospitalized without COVID-19 |
| Taha et al. 2020 | Sudan | Case-Control | 557/1000 | 1422/135 | (26–35): 41.8% CO | 42 CO | Patients with COVID-19 confirmed by RT-PCR | Healthy population |
| Solmaz et al. 2021 | Turkey | Cross-Sectional | 1667/ 127091 | 113868/ 14980 | ND | ND | Patients with COVID-19 confirmed by RT-PCR | Healthy population |
| Ad'hiah et al. 2020 | Iraq | Case-Control | 300/595 | ND | Mean Age ±SD: (49.7 ±12.3/29.3 ±6.9) | 59.7/49.7 | Patients with COVID-19 confirmed by RT-PCR | Healthy blood donors |
| Hoiland et al. 2020 | Canada | Retrospective Cohort | 95/398671 95/62246 | ND | Median Age (IQR) of 60%: 66 (58–73) CO | 64.2 CO | Patients with COVID-19 confirmed by RT-PCR | Healthy blood donors |
| Göker et al. 2020 [15] | Turkey | Retrospective Case-Control | 186/1882 | 1868/200 | Median Age (IQR): 42 (19–92) CO | 53.8 CO | Patients with COVID-19 confirmed by RT-PCR | Healthy blood donors |

## Discussion

The aim of the study was to assess the relationship between COVID-19 infection and different blood groups, as well as Rhesus state, using a meta-analysis method. Twenty-two studies were selected for blood type and ten for the Rhesus factor. Our results revealed that the blood groups A, B and AB are associated with an increase in the risk of COVID-19 infection in comparison with the O blood group, which seems to be protective. A mild publication bias was observed for the A and O blood group pair, through the visual inspection of the funnel plots and the results of Egger's test. Further, moderate to substantial heterogeneity, has been observed for the blood groups A and AB in comparison with the O blood group. Blood group B was characterized by the absence of heterogeneity.

Although the mechanisms that can explain the observed data have not yet been clarified, some assumptions can be made. The main one assumes that the anti-A and anti-B natural antibodies being produced in individuals with blood group O could potentially block viral

**Table 2. Meta-analysis results.**

| Blood groups / Rhesus status | Comparison | OR | 95% CI | 95% PI | I2 | 95% CI |
|---|---|---|---|---|---|---|
| ABO | A—AB | 0.98 | (082 to 117) | (048 to 198) | 0.25 | (0% to 56%) |
| | A—B | 1.1 | (098 to 123) | (067 to 179) | 0.26 | (0% to 56%) |
| | A—O | 1.29 | (115 to 144) | (079 to 21) | 0.54 | (25% to 71%) |
| | AB—B | 1.11 | (096 to 127) | (066 to 186) | 0.03 | (0% to 48%) |
| | AB—O | 1.32 | (110 to 157) | (067 to 259) | 0.41 | (2% to 65%) |
| | B–O | 1.15 | (106 to 125) | (087 to 153) | 0 | (0% to 38%) |
| Rhesus | Rh+ vs. Rh- | 0.97 | (083 to 113) | (061 to 154) | 0.38 | (0% to 70%) |

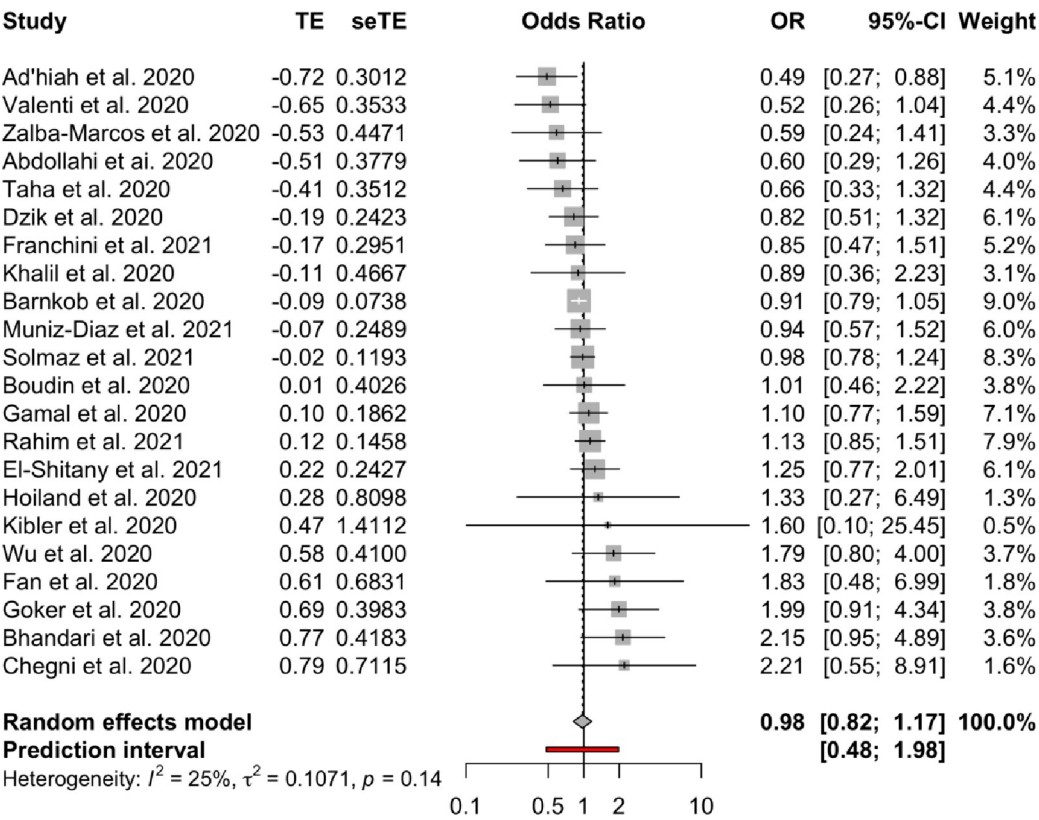

**Fig 2. Forest plots for the ABO gene comparison of A vs. AB group.**

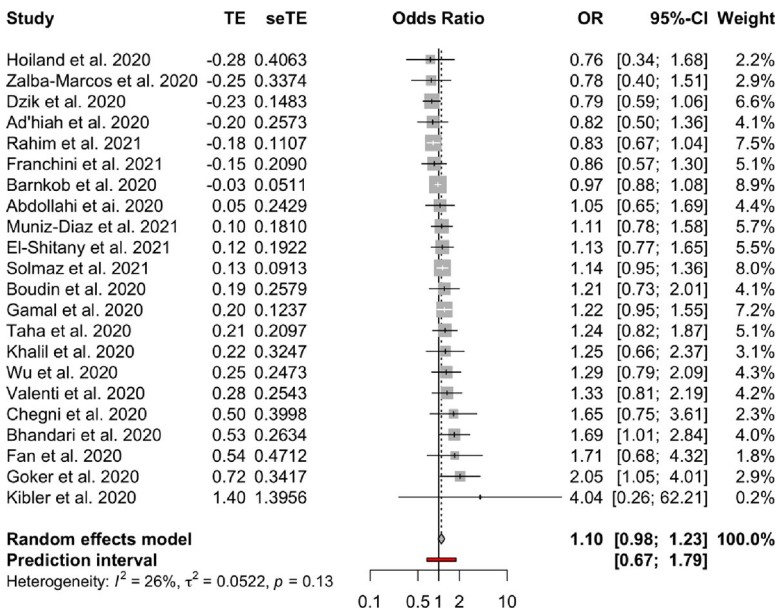

**Fig 3. Forest plots for the ABO gene comparison of A vs. B group.**

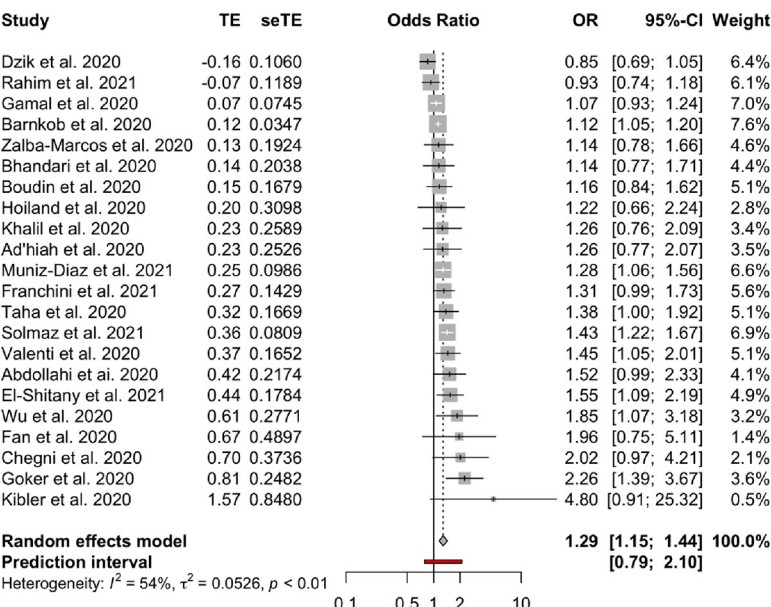

**Fig 4. Forest plots for the ABO gene comparison of A vs. O group.**

adhesion to cells, which could explain a lower risk of infection. Potential lack of such antibodies in blood groups A and B may explain the higher risk of COVID-19 infection but further studies are needed to elucidate this hypothesis [37]. Concerning the Rhesus status, there was not evidence of an association with COVID-19 infection. The visual inspection of the Rhesus factor funnel plot and the results of Egger's test showed moderate heterogeneity but no evidence of publication bias.

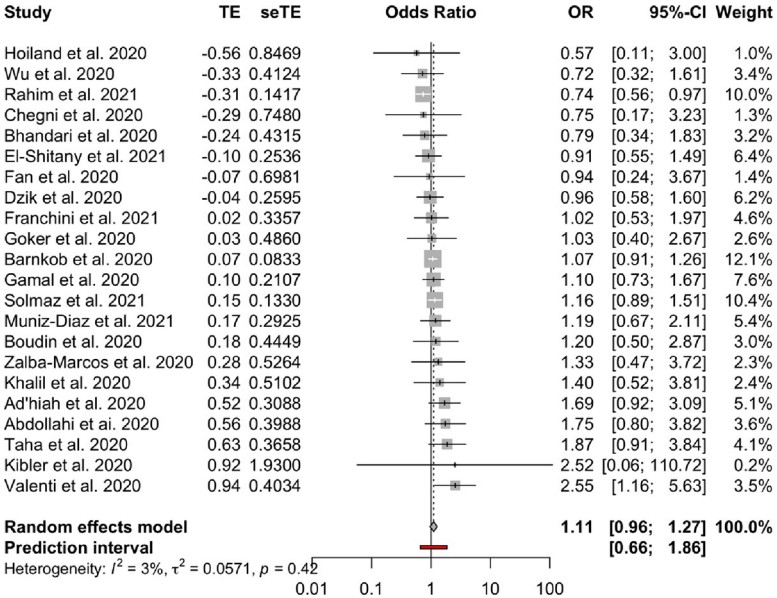

**Fig 5. Forest plots for the ABO gene comparison of B vs. AB group.**

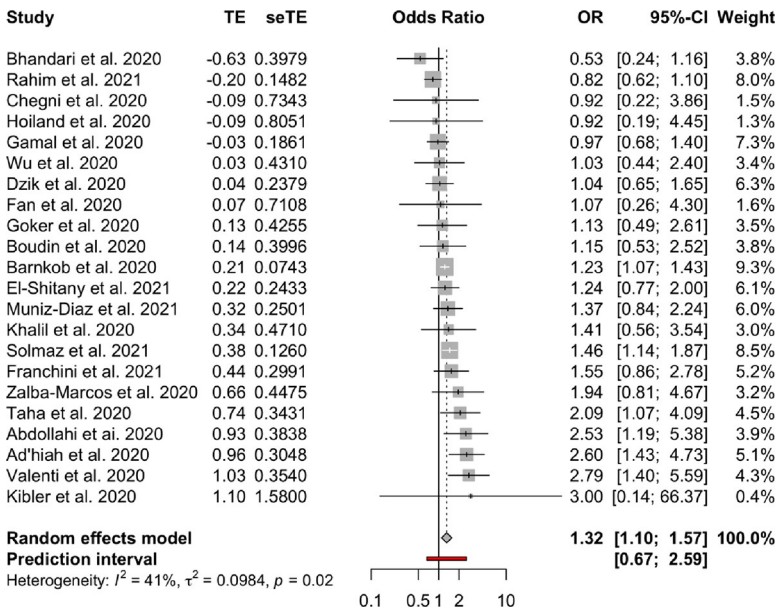

**Fig 6. Forest plots for the ABO gene comparison of O vs. AB group.**

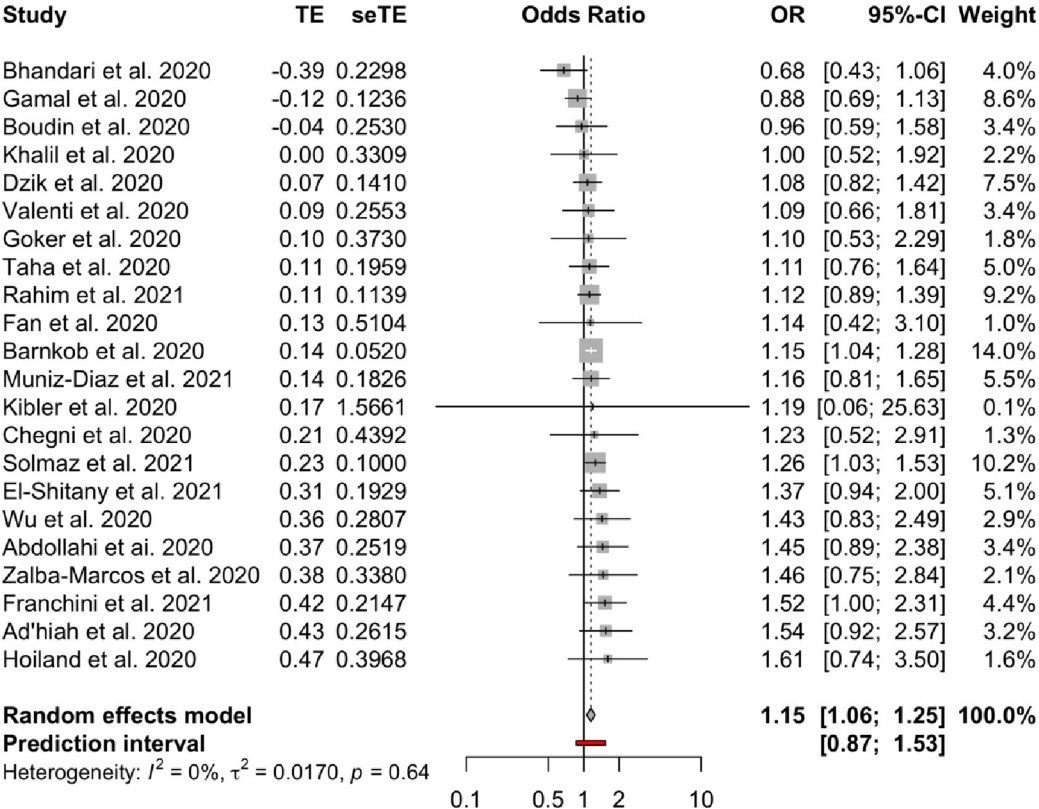

**Fig 7. Forest plots for the ABO gene comparison of B vs. O group.**

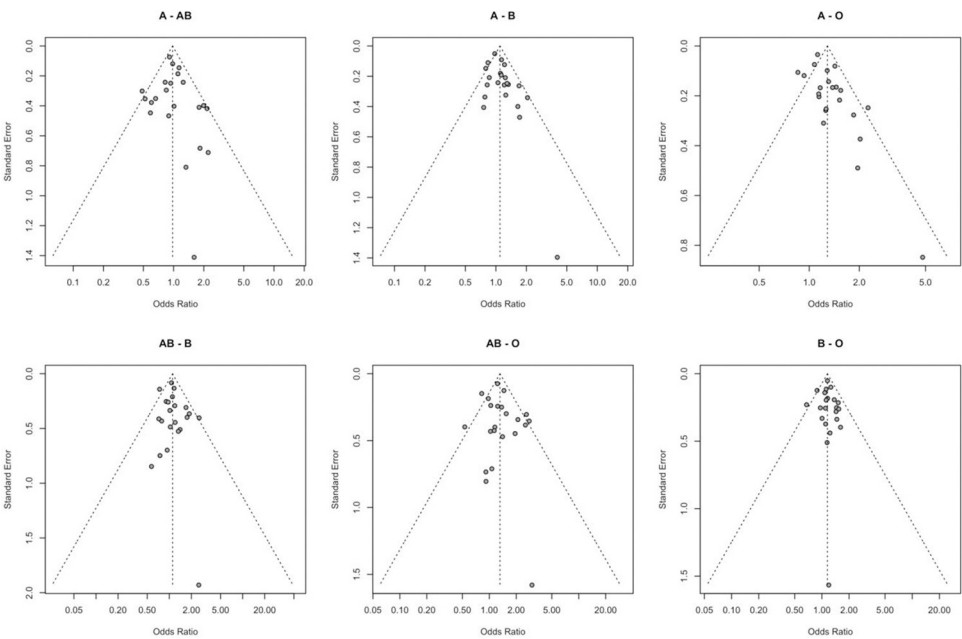

**Fig 8. Funnel plots for the ABO gene.**

The interpretation of the overall estimates should be done with caution because of the observed heterogeneity between studies. There was variability in the design and sample size, while a considerable part of the pooled control population comes mainly from a single study [38]. Further, the COVID-19 confirmation method was either genetic, clinical, or even unreported while potential confounding factors such as age, gender, race, region, and underlying diseases that may influence the predisposition to COVID-19 infection could not be accounted for due to absence of relevant information. Finally, the observed publication bias may be due to the study language chosen, which may have led to the exclusion of other relevant studies, in other languages [9]. Nevertheless, despite the unexplained heterogeneity, subgroup and sensitivity analysis still confirmed our results.

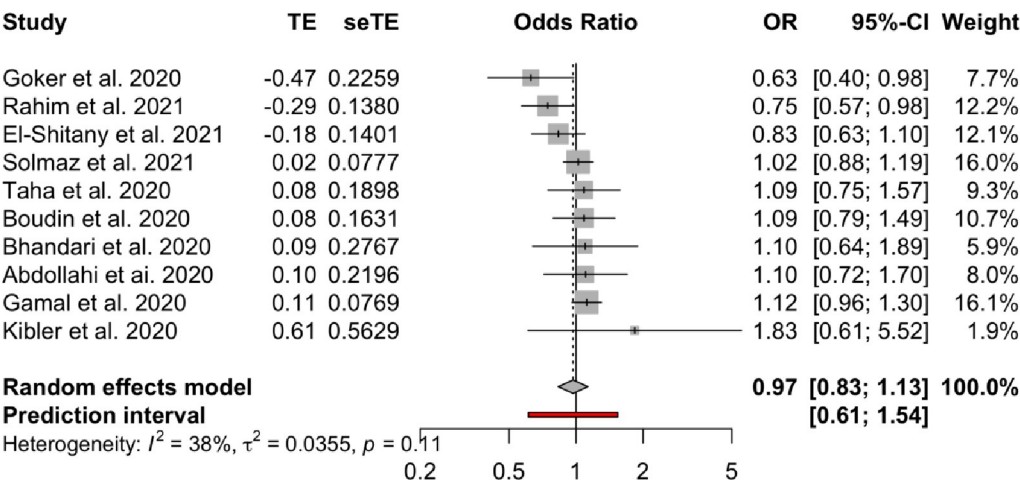

**Fig 9. Forest plot for the Rhesus status.**

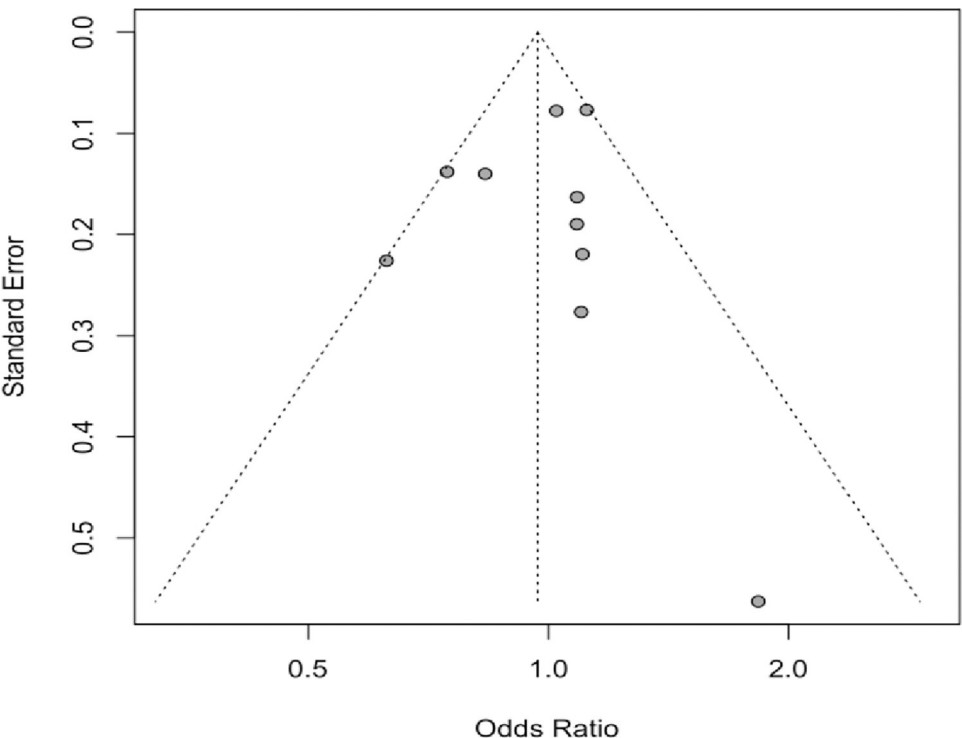

**Fig 10. Funnel plot for the Rhesus status.**

In conclusion, this meta-analysis provides evidence for an increased risk of COVID-19 infection for blood groups A, B and AB compared to blood group O, while an association between Rhesus state and COVID-19 infection could not be established.

## Supporting information

**S1 Checklist. PRISMA 2020 checklist.**
(PDF)

**S1 Table. Leave-one-out method results for ABO blood group.**
(XLSX)

**S2 Table. Leave-one-out method results for Rhesus.**
(XLSX)

## Author Contributions

**Conceptualization:** George Balaouras, Polychronis Kostoulas.

**Data curation:** George Balaouras, Paolo Eusebi.

**Formal analysis:** Paolo Eusebi.

**Funding acquisition:** Polychronis Kostoulas.

**Methodology:** George Balaouras, Paolo Eusebi.

**Project administration:** Polychronis Kostoulas.

**Resources:** Polychronis Kostoulas.

**Software:** Paolo Eusebi.

**Supervision:** Polychronis Kostoulas.

**Visualization:** George Balaouras.

**Writing – original draft:** George Balaouras.

**Writing – review & editing:** Paolo Eusebi, Polychronis Kostoulas.

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
