## [Decision Letter · Decision Letter 0]

6 Nov 2021

PONE-D-21-27109Systematic review and meta-analysis of the effect of ABO blood group on the risk of COVID-19 infectionPLOS ONE

Dear Dr. Kostoulas,

Thank you for submitting your manuscript to PLOS ONE. After careful consideration, we feel that it has merit but does not fully meet PLOS ONE’s publication criteria as it currently stands. Therefore, we invite you to submit a revised version of the manuscript that addresses the points raised during the review process.

The work is certainly of interest, and the Authors should be commended for their effort. However, given the importance of the topic, some issues - listed below - deserve further attention, mainly pertaining the searching time period, the inclusion criteria, finally the assessment of COVID-19, as reported in the included studies. In addition to the issues listed below, please acknowledge that the terminology used (and, in turn, the main aim of the study) is somehow misleading: if the present meta-analysis is focused on the association between blood type and the likelihood of infection, then you should not use the expression "COVID-19 infection", as COVID-19 refers to the disease - which may arise in a subset of patients infected by SARS-CoV-2 - but "SARS-CoV-2 infection". Conversely, if the present meta-analysis is focusing on the association between blood type and the likelihood of the disease (i.e. COVID-19), the term "infection" is redundant. But, in the latter case, all the included studies assessing the presence of the virus through RT-PCR, regardless of the clinical status of the patients, should be properly re-screened, to include only those with symptomatic subjects (if we are here evaluating the likelihood of the disease onset). Finally, if we are trying to quantify the association between blood type and both infection and/or disease onset, these are clearly two separate analyses, and the studies included in each one should be properly selected. In this regard, please consider also point 3 raised by the Reviewer 3. 

We look forward to receiving your revised manuscript.

Kind regards,

Maria Elena Flacco, M.D.

Academic Editor

PLOS ONE

“Funding unde the H2020 project:

Unravelling Data for Rapid Evidence-Based Response to COVID-19”

Reviewers' comments:

Reviewer's Responses to Questions

**Comments to the Author**

1. Is the manuscript technically sound, and do the data support the conclusions?

Reviewer #1: Partly

Reviewer #2: Yes

Reviewer #3: No

2. Has the statistical analysis been performed appropriately and rigorously? 

Reviewer #1: Yes

Reviewer #2: Yes

Reviewer #3: No

3. Have the authors made all data underlying the findings in their manuscript fully available?

Reviewer #1: Yes

Reviewer #2: Yes

Reviewer #3: Yes

4. Is the manuscript presented in an intelligible fashion and written in standard English?

Reviewer #1: Yes

Reviewer #2: Yes

Reviewer #3: Yes

5. Review Comments to the Author

Reviewer #1: At least a systematic review on this topic has been already published, with similar conclusions and better methodologic assessment of the included studies. Since then, no news evidence has become available. Therefore I dont think that presently a replication of a previous systematic review is likely to have value.

Reviewer #3: 

- The abstract should include the statistically significant, import effect estimates.

- “The searching time period was restricted between February 1st 2021 to March 7th 2021”. Why are these search dates so restrictive? You are only looking at studies published within 1 month of each other? This does not make sense.

- Studies were included if Covid-19 was “clinically diagnosed”. This seriously undermines the scientific usefulness of this work. Pure clinical diagnosis of Covid-19 is very difficult given the range of presentations. Confirmatory testing is almost always used. Please exclude studies that relied on “clinically diagnosed” Covid-19.

- This study misses many reports of the association of blood type and Covid-19, but it is unclear why these studies were omitted: Ellinghaus et al., Kolin et al., Zietz et al., etc.

- The reason for excluding particular reports, along with the number of reports excluded for that particular reason, needs to be listed in Figure 1.

6. PLOS authors have the option to publish the peer review history of their article (what does this mean?). If published, this will include your full peer review and any attached files.

Reviewer #1: No

Reviewer #2: **Yes: **Eduardo Muñiz-Diaz

Reviewer #3: No

---

## [Author Response · Author response to Decision Letter 0]

15 Apr 2022

A detailed response to the reviewers' comments has been attached as a separate file.

---

## [Decision Letter · Decision Letter 1]

24 Jun 2022

PONE-D-21-27109R1Systematic review and meta-analysis of the effect of ABO blood group on the risk of SARS-CoV-2 infectionPLOS ONE

Dear Dr. Kostoulas,

Thank you for submitting your manuscript to PLOS ONE. After careful consideration, we feel that it has merit but does not fully meet PLOS ONE’s publication criteria as it currently stands. Therefore, we invite you to submit a revised version of the manuscript that addresses the points raised during the review process.

ACADEMIC EDITOR: Thank you for sending us your manuscript.

Take into account the only suggestions of the reviewer and your manuscript will be ready to be published.

We look forward to receiving your revised manuscript.

Kind regards,

Kovy Arteaga-Livias

Academic Editor

PLOS ONE

Journal Requirements:

Reviewers' comments:

Reviewer's Responses to Questions

**Comments to the Author**

1. If the authors have adequately addressed your comments raised in a previous round of review and you feel that this manuscript is now acceptable for publication, you may indicate that here to bypass the “Comments to the Author” section, enter your conflict of interest statement in the “Confidential to Editor” section, and submit your "Accept" recommendation.

Reviewer #2: All comments have been addressed

Reviewer #4: All comments have been addressed

2. Is the manuscript technically sound, and do the data support the conclusions?

Reviewer #2: (No Response)

Reviewer #4: Yes

3. Has the statistical analysis been performed appropriately and rigorously? 

Reviewer #2: (No Response)

Reviewer #4: Yes

4. Have the authors made all data underlying the findings in their manuscript fully available?

Reviewer #2: (No Response)

Reviewer #4: Yes

5. Is the manuscript presented in an intelligible fashion and written in standard English?

Reviewer #2: (No Response)

Reviewer #4: Yes

6. Review Comments to the Author

Reviewer #2: (No Response)

Reviewer #4: Excellent work, the only suggestion I have, is that you should add in your table, the characteristics of the confounding or adjustment variables in each study. Also, the adjustment methods used in these studies.

7. PLOS authors have the option to publish the peer review history of their article (what does this mean?). If published, this will include your full peer review and any attached files.

Reviewer #2: No

Reviewer #4: **Yes: **Joshuan J. Barboza

---

## [Author Response · Author response to Decision Letter 1]

29 Jun 2022

Dear Editor and Reviewers

Thank you for being positive to the revised version that we submitted. There is only one suggestion raised by R4 which we decided not to follow, and we explain in our response to reviewers why.

We would like to thank the Editor and Reviewers for their suggestions and for being positive on the revised version that we submitted.

---

## [Editor Report · Decision Letter 2]

1 Jul 2022

Systematic review and meta-analysis of the effect of ABO blood group on the risk of SARS-CoV-2 infection

PONE-D-21-27109R2

Dear Dr. Kostoulas,

We’re pleased to inform you that your manuscript has been judged scientifically suitable for publication and will be formally accepted for publication once it meets all outstanding technical requirements.

Kind regards,

Kovy Arteaga-Livias

Academic Editor

PLOS ONE
---

## [Editor Report · Acceptance letter]

6 Jul 2022

PONE-D-21-27109R2 

Systematic review and meta-analysis of the effect of ABO blood group on the risk of SARS-CoV-2 infection 

Dear Dr. Kostoulas:

I'm pleased to inform you that your manuscript has been deemed suitable for publication in PLOS ONE. Congratulations! Your manuscript is now with our production department. 

Kind regards, 

on behalf of

Dr. Kovy Arteaga-Livias 

Academic Editor

PLOS ONE